# Endothelial-Tumor Cell Interaction in Brain and CNS Malignancies

**DOI:** 10.3390/ijms21197371

**Published:** 2020-10-06

**Authors:** Maria Peleli, Aristidis Moustakas, Andreas Papapetropoulos

**Affiliations:** 1Department of Medical Biochemistry and Microbiology, Science for Life Laboratory, Uppsala University, Box 582, SE-751 23 Uppsala, Sweden; aris.moustakas@imbim.uu.se; 2Clinical, Experimental Surgery and Translational Research Center, Biomedical Research Foundation of the Academy of Athens, 115 27 Athens, Greece; apapapet@pharm.uoa.gr; 3Laboratory of Pharmacology, Faculty of Pharmacy, National and Kapodistrian University of Athens, 157 71 Athens, Greece

**Keywords:** glioblastoma, brain tumors, endothelium, angiogenesis, VEGF, TGF-β, reactive oxygen species, nitric oxide, gap junctions

## Abstract

Glioblastoma and other brain or CNS malignancies (like neuroblastoma and medulloblastoma) are difficult to treat and are characterized by excessive vascularization that favors further tumor growth. Since the mean overall survival of these types of diseases is low, the finding of new therapeutic approaches is imperative. In this review, we discuss the importance of the interaction between the endothelium and the tumor cells in brain and CNS malignancies. The different mechanisms of formation of new vessels that supply the tumor with nutrients are discussed. We also describe how the tumor cells (TC) alter the endothelial cell (EC) physiology in a way that favors tumorigenesis. In particular, mechanisms of EC–TC interaction are described such as (a) communication using secreted growth factors (i.e., VEGF, TGF-β), (b) intercellular communication through gap junctions (i.e., Cx43), and (c) indirect interaction via intermediate cell types (pericytes, astrocytes, neurons, and immune cells). At the signaling level, we outline the role of important mediators, like the gasotransmitter nitric oxide and different types of reactive oxygen species and the systems producing them. Finally, we briefly discuss the current antiangiogenic therapies used against brain and CNS tumors and the potential of new pharmacological interventions that target the EC–TC interaction.

## 1. Brain and CNS Malignancies

Malignant tumors of the central nervous system (CNS) are among the malignancies with the poorest prognosis. This is indicated by the association of brain tumors with the highest estimated number of years (mean: ~20 years) of potential life lost due to any type of tumor [1]. Below, we briefly describe the main types of brain and CNS malignancies such as gliomas, medulloblastomas, and neuroblastomas.

### 1.1. Gliomas

Gliomas are primary brain tumors, mostly found in adults, that are thought to derive from neuroglial stem or progenitor cells. On the basis of their histological appearance, they have been traditionally classified as astrocytic, oligodendroglial, or ependymal tumors and assigned WHO grades I–IV, which indicate different degrees of malignancy [2]. The WHO grading system reflects tumor malignancy and presumed natural disease course, with WHO grade I indicating a slow-growing lesion usually associated with favorable prognosis, whereas WHO grade IV is assigned to highly malignant tumors [3]. Glioblastoma accounts for 82% of cases of malignant glioma usually derived from astrocytes or oligodendrocytes and is characterized histologically by considerable cellularity and mitotic activity, vascular proliferation, and necrosis [4]. Because cells in these tumors vary in size and shape, i.e., they are pleomorphic, glioblastomas were called glioblastoma multiforme, a term no longer in use. Glioblastoma and other malignant gliomas are highly invasive, infiltrating surrounding brain parenchyma, yet they are typically confined to the central nervous system (CNS) and only in very rare cases metastasize outside the CNS, as, for example, metastases to the skull bone have been observed [5].

### 1.2. Medulloblastomas

Medulloblastoma (MB) is the most common malignant brain tumor in children. Despite aggressive therapies, approximately one-third of MB patients die from the disease and those who survive often suffer severe side effects from therapy, including cognitive deficits, endocrine disorders, and secondary tumors [6].

Medulloblastoma (MB) comprises a biologically heterogeneous group of embryonal tumors of the cerebellum. Four subgroups of MB have been described (WNT, Sonic Hedgehog (SHH), Group 3, and Group 4), each of which is associated with different genetic alterations, age at onset, and prognosis [7]. The origin of the medulloblastoma stem cells which eventually gives rise to the tumor remains still controversial. However, mutations activating Wnt have been identified primarily in the classic medulloblastoma subtype, which is thought to derive primarily from stem and/or progenitor cells of the ventricular zone in the cerebellum. The external germinal layer (EGL) of the cerebellum, in contrast, is thought to give rise to nodular/desmoplastic medulloblastoma commonly associated with activation of the SHH pathway. Finally, other cerebellar stem and progenitor cells have been described and may also give rise to medulloblastoma [8].

### 1.3. Neuroblastomas

Neuroblastoma accounts for more than 7% of malignancies in patients younger than 15 years and around 15% of all pediatric oncology deaths [9]. It is the most common extracranial solid tumor in childhood and the most frequently diagnosed neoplasm during infancy [10]. Neuroblastoma is a disease of the sympaticoadrenal lineage of the neural crest, and therefore tumors can develop anywhere in the sympathetic nervous system. Regarding the exact origin of neuroblastoma cells, it is widely accepted that they originate from neural crest cells (NCC). NCC is a group of embryonic cells located in proximity to the neural tube. During embryonic development, NCC migrate to generate the ganglia of sympathetic nervous system and the adrenal medulla [11]. Most primary tumors (65%) occur within the abdomen, with at least half of these arising in the adrenal medulla. Other common sites of disease include the neck, chest, and pelvis [12].

## 2. Mechanisms of Formation of New Vessels that Promote Further Tumorigenesis in Brain and CNS Tumors

Malignant cells require oxygen and nutrients in order to survive and proliferate, and thus they need to be in close proximity to blood vessels, which mediate molecular transport to every tissue [13]. High microvessel density correlates with tumor growth and is an independent marker of poor patient prognosis [14]. Tumors achieve their vascularization by inducing new blood vessel formation, generally termed as tumor angiogenesis, or by relying on pre-existing vasculature, a phenomenon described as vascular co-option [15].

A blood vessel consists of different layers and cell types, but among them, endothelial cells (EC) are the most critical since they form capillaries which supply the tissues with oxygen and nutrients and remove waste products [16]. In addition, EC are essential cellular building blocks that contribute to the development and homeostasis of the vascular system and impact on tissue architecture and activity [17]. Normally, ECs in adult organisms remain quiescent unless activated to proliferate by pro-angiogenic signals, of which vascular endothelial growth factor (VEGF) is the prototype [18,19]. This transition of EC from quiescence to an activated, proliferative state is termed “angiogenic switch” during which ECs differentiate into distinct subpopulations to serve different functions [20]. The angiogenic switch critically releases tumors from a relative state of dormancy to a more proliferative and aggressive phenotype which is in close relation with the formation of new vessels [13]. A wide range of mechanisms have been described that can initiate new vessel formation in tumors. Below, we describe some important mechanisms in relation to brain and CNS tumors (summarized in Figure 1).

### 2.1. Vasculogenesis and Recruitment of EC Progenitors from the Bone Marrow

Vasculogenesis refers to the de novo formation of new blood vessels by colonization of circulating EC or by endothelial progenitor cells (EPCs) derived from the bone marrow [13]. Vasculogenesis happens physiologically during embryogenesis or after an ischemic insult [21,22], but it also characterizes many cancer types where it favors tumor progression [23].

In a hallmark study, Kion et al. showed that inhibition of vasculogenesis, but not sprouting angiogenesis, prevents the recurrence of glioblastoma after irradiation in mice [24]. In that case, vasculogenesis was mediated mainly by the recruitment of EPCs from the bone marrow and the interaction between stromal derived factor 1 (SDF-1) and its receptor CXCR4. SDF-1 was secreted by glioblastoma cells and served as a chemoattractant for EPCs that expressed the receptor CXCR4 [24]. The importance of SDF-1 on de novo vasculogenesis was further confirmed in a subsequent study of a murine astrocytoma model [25]. It is worth mentioning that vasculogenesis from bone marrow-derived cells had been also shown earlier in a study where the investigators used a mouse model of mammary carcinoma metastasis to the brain [26]. Based on these preclinical studies, a clinical phase I/II study was published in 2019, showing that the infusion of the CXCR4 inhibitor plerifaxor was well tolerated as an adjunct to standard chemoirradiation with newly diagnosed glioblastoma and improved local control of tumor recurrences [27]. The results from this clinical trial are encouraging; however, since this was a small and non-randomized study, future larger randomized trials are warranted.

Finally, when it comes to pediatric tumors, it has been shown that high levels of circulating VEGFR2^+^ bone marrow-derived progenitor cells correlate with metastatic disease in children with solid malignancies, including brain tumors such as medulloblastoma, brain stem glioma, and brain stem pilocytic astrocytoma [28].

### 2.2. Sprouting Angiogenesis

EC at the leading edge of growing vascular networks are called tip cells and sense molecular queues that direct them to nearby areas with high abundance of pro-migratory growth factors [29,30]. On the other hand, stalk and phalanx EC trail behind the tip EC, can generate sprouts, and support the newly formed vessels both physically and via their cell divisions (Figure 1). During sprouting angiogenesis, VEGF and Notch signaling pathways specify the fate of the tip and stalk EC [31,32]. A VEGF gradient, often derived from the tumor cells, guides the tip EC and determines the direction and formation of new vessels [33]. On the contrary, Delta-like canonical Notch ligand 4 (DLL4) expressed on the tip EC binds to the Notch receptor on the stalk EC and inhibits angiogenesis by exerting an inhibitory effect on the local VEGF signaling [34]. The interdependence between Notch and VEGF signaling is probably a universal mechanism in angiogenesis with the Notch attenuating the action of VEGF and vice versa. Thus, in the context of cancer, VEGF promotes sprouting angiogenesis and tumor growth, whereas Notch has an opposite inhibitory effect [35].

Sprouting angiogenesis is also abundant in the brain and CNS and seems to play a considerable role in respective tumors. In particular, analyses on glioblastomas have shown that a considerable number of vascular EC undergo rapid cell divisions and probably represent stalk cells [36,37]. Moreover, in situ analysis in glioma biopsies indicates the presence of tip cells, based on the expression of many of their marker proteins, such as VEGF receptor 2 (VEGFR2), neuropilin-1, angiopoietin-2, and integrin-β1 [38]. In addition, the DLL4-Notch signaling system is present in the CNS during physiological retinal angiogenesis and restricts the number of tip cells formed after VEGF treatment, and therefore limits the extent of sprouting angiogenesis [34]. The same effect of DLL4-Notch signaling has been also observed in gliomas where it was shown that inhibiting the DLL4-Notch signaling pathway increases the degree of vessel sprouting inside the tumor [39]. However, and in contrast to what has been observed in other cancer types, this increase of vessel sprouting was associated with a surprising restriction on tumor growth.

All in all, sprouting angiogenesis does occur in the normal adult brain and in brain tumors with VEGF and Notch signaling playing key regulatory roles. However, the exact effect that sprouting angiogenesis has on tumor growth remains controversial and possibly exhibits disease type specificity.

### 2.3. Intussusceptive Angiogenesis

Intussusception represents another, albeit less frequently studied, type of angiogenesis. Intussusceptive angiogenesis is a physiological process that engages pre-existing vessels, formed either by early embryonic vasculogenesis or sprouting angiogenesis during organismic development [40]. During intussusceptive angiogenesis, preexisting vessels split into daughter vessels, creating a more complicated vascular network that can better supply the developing tissue, even the tumor, with nutrients and oxygen.

In detail, transluminal tissue pillars are formed within pre-existing vessels which then fuse, thus delineating new vascular entities or resulting in vessel remodeling [41]. Similar to sprouting angiogenesis, VEGF signaling plays a key role in the induction of intussusceptive angiogenesis. Other important growth factors in this process are platelet-derived growth factor (PDGF) and erythropoietin [42,43,44]. Furthermore, Notch1 signaling plays a suppressive role in both physiological and cancer intussusceptive angiogenesis [45,46]. However, the molecular and cell-based mechanisms are less well understood compared to sprouting angiogenesis.

Among many different types of cancers, intussusceptive angiogenesis has been described in brain tumors [47]. Grade II gliomas show increased microvascular density of well-sealed vessels which are generated via vessel intussusception and vessel co-option mechanisms. In grade III and especially in grade IV gliomas, as the size of the tumor and the level of hypoxia increases, there is a pressure to generate many newly formed vessels, and in that case, the angiogenic switch described earlier takes place [48]. Finally, it is worth mentioning that preclinical mouse models have shown that intussusceptive angiogenesis might contribute to angiogenesis when tumors develop resistance to traditional antiangiogenic agents, which are usually combined with standard radio-/chemo-therapeutic treatment. In particular, mice exhibited an increase in CD34^+^ EC (marker for intussusception) in an orthotopic U87MG mouse model of grade IV glioma at day 6 after initiation of the treatment [37].

### 2.4. Vasculogenic Mimicry and Transdifferentiation of Cancer Cells

Vasculogenic mimicry (VM) is a phenomenon whereby the tumor cells themselves form “vessel-like structures” independent from angiogenesis based on EC, thus generating a system that supplies the tumor with nutrients and oxygen [49]. VM takes place in many cancer types, including brain tumors, and it is associated with poor prognosis for the patient [13]. Structures formed through the process of VM are identified using EC protein markers such as CD31 and VE-Cadherin [50,51,52]. Despite the expression of these markers, it should be emphasized that there is a lack of technology that could be used to unambiguously distinguish VM from normal EC lining which makes the investigation of this process difficult [13].

Regarding VM in brain tumors, it has been reported in gliomas via glioblastoma stem cells that trans-differentiate to EC and in neuroblastomas and in medulloblastomas after antiangiogenic therapy [53,54,55,56]. There are several molecular mediators of VM in gliomas, which is the most studied brain tumor. For example, tumor associated macrophages (TAMs) seem to promote VM through cyclooxygenase-2 (COX-2) activation [57]. Moreover, the specific expression of VE-Cadherin (CDH5) in glioblastoma stem cells may contribute to vasculogenesis, especially under hypoxia where VE-Cadherin gene expression is upregulated, when hypoxia-inducible factor-1 or -2 (HIF-1 or -2) bind to and transactivate a VE-Cadherin promoter [58]. Finally, a recent study showed that the phosphatidyl-inositol 3′ kinase (PI3K)/Akt/mammalian target of rapamycin (mTOR) pathway is involved in inducing VM in gliomas [59]. When it comes to medulloblastomas, the literature is more limited; however, a study indicated that VM was present in about 22% of the examined medulloblastoma tissues and was associated with higher expression of matrix metalloproteinases (MMP-2, MMP-14), ephrin (Eph) A2, and laminin 5γ2. Importantly, the presence of VM was associated with lower survival and poor prognosis [56]. Regarding neuroblastoma, VM mimicry is highly associated with MYNC positive tumors [12,54]. MYNC (N-Myc) is an oncogene that encodes a transcription factor that belongs to the MYC family. Normally, MYNC is thought to be critical in brain development [60]. However, MYNC amplification is found in 20–30% of neuroblastoma tumors and is associated with more aggressive features of the cancer at the time of diagnosis [61,62]. In neuroblastoma, “EC-like” cells have been found which are positive for EC protein markers such as CD31, CD105, and VEGFR2. VEFGR2 is highly expressed in neuroblastoma cells, and this is believed to play a critical role in advancing tumor progression [54].

A process similar to VM is the trans-differentiation of cancer stem cells to EC, which in turn gives rise to neovascularization. In trans-differentiation, the expression of differentiation markers that are typical of EC is higher compared to their expression in cells generated via VM [63]. Trans-differentiation has been shown in vitro, but its presence in vivo is highly questionable in the context of brain tumors [13]. In particular, in vitro studies have shown that glioblastoma stem cells in endothelial cell promoting media can express EC markers like CD31, CD34, and von Willebrand factor, and they can form tubular structures [53,64]. However, these studies have not been confirmed in vivo or in human glioblastoma biopsies and it is actually more likely that glioblastoma stem cells can differentiate in vivo into pericytes, which play a critical role in tumor vessel formation and tumor growth [65].

### 2.5. Vascular Co-Option

Vascular co-option is another key mechanism used by cancer cell types in order to gain access to nutrients and oxygen. Similar to vascular mimicry described above, here again, the most important role is played by the tumor cells, and therefore this process is considered largely independent of angiogenesis [66]. During this process, the cancer cells move towards pre-existing vessels, and thus not only gain access to nutrients and oxygen but also further invade inside the surrounding normal tissues [67]. Vessel co-option has been so far mostly described in highly vascularized tumors such as brain, lung, and liver, and it can explain to some extent the antiangiogenic drug resistance observed in many tumors [66].

Regarding brain tumors, co-option has been extensively studied in glioblastoma, but it has also been reported in animal models of medulloblastoma and neuroblastoma after anti-VEGF therapy [67,68,69,70]. Moreover, vascular co-option has been documented in cases of brain tumors derived after metastasis of other cancer types such as breast cancer or melanoma in rodent models, or even from human clinical specimens where the primary tumor was breast cancer, melanoma, or lung cancer [71,72,73].

When it comes to the molecular players driving vascular co-option in brain tumors, most of the studies have focused on glioblastoma so far; some experimental evidence in medulloblastoma also exists. In glioblastoma, important molecular modulators include: (a) bradykinin (which belongs to the kinin family), which is secreted by EC and induces chemotaxis of the glioblastoma cells; (b) the SDF1/CXCR4 chemoattractant system which was described earlier in the vasculogenesis section; (c) interleukin (IL)-8 secreted by the EC; (d) mutated versions of epidermal growth factor receptor (EGFR) expressed in glioblastoma cells such as the EGFRvIII, which contributes to excessive angiogenesis; (e) the factor MDGI (mammary-derived growth inhibitor), which plays a key role in maintaining lysosomal membrane stability and is overexpressed in both glioblastoma cells and the EC; and (f) EphrinB2, which is of EC origin and plays a fundamental role in cell-to-cell communication [74,75,76,77,78,79]. Blockade of or interference with the above-mentioned pathways has been shown to successfully reduce the levels of vascular co-option in glioblastoma animal models. Further studying of the molecular mechanisms involved in vascular co-option in glioblastoma is of great clinical importance since some of these pathways can be pharmacologically inhibited by drugs that are already approved for clinical use, and vascular co-option seems to be one of the major mechanisms used by the glioblastoma cells for escaping antiangiogenic therapy [66,76].

Regarding pediatric brain tumors, it has been described that loss of the tumor suppressor gene PTEN in medulloblastoma creates an abnormal perivascular niche of tumor cells [70]. In agreement with loss of PTEN, the PI3K/Akt pathway might play a critical role in medulloblastoma by increasing the survival of cancer stem cells residing in the perivascular niche following radiation therapy [69]. It is worth mentioning that these in vivo animal studies in medulloblastoma studied the behavior of the cancer cells in the perivascular niche and not vascular co-option per se, and thus the importance of the co-option mechanism in medulloblastoma warrants further investigation.

### 2.6. Role of EMT and EndoMT in Brain Tumors

The epithelial-to-mesenchymal transition (EMT) is a trans-differentiation process where epithelial cells gradually lose some epithelial markers, loosen their cell–cell junctions, and obtain more mesenchymal characteristics that favor cell motility [80]. This is a physiological response under circumstances when cells need to invade and occupy empty space like in embryonic development or in wound closure; however, EMT can also play a significant pathological role in cancer since EMT can facilitate tumor cell invasiveness and metastasis [81]. EMT has been shown to take place in many animal models of cancer, but direct evidence that EMT is of relevance in human cancers has been more difficult to obtain, possibly due to the fact that EMT is a transient phenomenon and there is still a lack of a good panel of markers that probe the EMT process and not the epithelial or mesenchymal end states [82].

EMT is triggered by TGF-β signaling which activates pathways including the Smad proteins and the mitogen-activated protein kinases. The activation of these pathways leads in turn to activation of transcription factors (i.e., Snail 1 and 2, ZEB 1 and 2, and Twist 1 and 2) which alter the gene expression in a way that suppresses the expression of epithelial-related genes (i.e., E-Cadherin) and induces the expression of mesenchymal-related genes (i.e., N-Cadherin) [83].

One key difference of gliomas and other brain tumors compared to solid epithelial tumors is the lack of a basement membrane and the very low expression of E-Cadherin [84]. However, similar phenomena to EMT do take place in brain tumors and help the cancer cells acquire stem-like properties [85]. In particular, it has been shown that brain tumor-derived mesenchymal cells are recruited during gliomagenesis in mice and contribute to further tumor progression [86]. Finally, it is worth mentioning that EMT may indirectly facilitate the growth and expansion of blood vessels around the tumor since EMT has been shown to favor phenomena such as vascular co-option and the vascular mimicry [87].

A similar event to EMT is the endothelial-to-mesenchymal transition (EndoMT), which involves the “transformation” of the endothelial cells to a more mesenchymal (less differentiated) phenotype [88]. Since the endothelial cells have a key role in the maintenance of functional capillaries, and thus the supply of tissues with oxygen and nutrients, a deregulation of the endothelium is a hallmark in many pathologies, e.g., cardiovascular diseases, tumor growth, and metastasis [89]. Although the knowledge regarding EndoMT is more limited, recent publications indicate that it can be triggered by known growth factors such as TGF-β and PDGF-B [90,91]. So far, many aspects of the role of EndoMT in cancer remain largely unknown, but two hallmark publications in 2018 suggested that EndoMT contributes to phenomena such as resistance to radiotherapy in lung cancer and resistance to conventional antiangiogenic therapies that target VEGFR2 in glioblastoma [91,92]. Therefore, finding new pharmacological agents that target EndoMT in glioblastoma might be a future promising approach for enhancing the efficacy of the current radiotherapies and antiangiogenic therapies.

## 3. Levels of EC–Tumor Cell Interaction

There are multiple levels of interaction between the endothelial and the tumor cells, which are described in detail below and summarized in Figure 2.

### 3.1. Interaction with Secreted Factors

#### 3.1.1. VEGF

VEGF is the most potent angiogenic factor that has been described until now, among hundreds of other pro-angiogenic molecules [93]. VEGF in brain tumors is both tumor- and EC-derived and therefore exerts its effects in a paracrine and autocrine fashion through high affinity binding to the tyrosine kinase receptors VEGFR2/Flk-1 and VEGFR1/Flt-1 [94,95]. VEGF has a plethora of biological functions as it induces angiogenesis, vasculogenesis, blood vessel permeability, and protein extravasation, which leads to vasogenic edema that is observed in the majority of brain tumors [96]. This edema dramatically increases the intracranial pressure by blood–brain barrier (BBB) leakage and it is responsible to a great extent for the high morbidity and mortality rates observed in glioma patients [97].

#### 3.1.2. FGF2

Fibroblast growth factors (FGFs) constitute a large family of growth factor polypeptides comprised of 19 members [93]. Among them, FGF2 seems to be the most important one when it comes to the regulation of angiogenesis in brain tumors [98]. FGF2 binds to FGFR1, which is a tyrosine kinase receptor mostly expressed on EC, and the downstream signaling favors angiogenesis in two ways: a) modulation of EC activity and (b) regulation of VEGF expression by the tumor cells [99]. In particular, FGF2 has been shown to upregulate urokinase plasminogen activator (uPA), its respective receptor, and also the expression of collagenases on the EC [100,101]. All these events apparently facilitate EC migration through the extracellular matrix and support tumor growth. Moreover, FGF2 has been shown to act as a chemotactic agent for EC and to help them form capillary-like tubes [102,103]. Finally, FGF2 has been proven to directly induce VEGF expression and production, which is, as discussed above, the master potentiator of angiogenesis in brain tumors [99].

#### 3.1.3. TGF-β

TGF-β is a growth factor affecting a plethora of cellular processes such as differentiation, adhesion, motility, proliferation, and apoptosis [104]. TGF-β binds to two different serine/threonine receptor kinases, type I and II, which activate in turn a family of signaling proteins called Smads [105]. It has been reported that glioblastoma cells secrete the active forms of TGF-β isoforms 1 and 2 but not TGF-β 3 [106]. Later on, it was shown that TGF-β 1 and 2 are strongly expressed in human gliomas and the expression levels correlate with the grade of malignancy [107]. Moreover, transcriptional profiling of human glioblastoma vessels indicated a key role of VEGF-A and TGF-β 2 in promoting vascular abnormalities in glioblastoma [108]. Finally, high TGF-β-induced Smad activity has been shown to confer poor prognosis in glioblastoma patients and to promote cell proliferation through the induction of PDGF-B in gliomas [109]. PDGF-B had been earlier shown to enhance glioma angiogenesis by stimulating VEGF expression in the tumor EC and by promoting pericyte recruitment in the tumor site [110].

#### 3.1.4. PDFG-B

PDGF-B is another growth factor expressed in human gliomas whose role in tumorigenesis has been shown to be more controversial and dependent on the cell of origin [111,112]. In particular, glioblastoma cell-derived PDGF-B enhances angiogenesis by stimulating VEGF production in EC overexpressing the PDGF receptor β [110]. However, the exact opposite effect has been shown when PDGF-B is overexpressed and derived from the vascular EC surrounding the tumor mass [113].

#### 3.1.5. Pleiotrophin

Pleiotrophin is a small heparin-binding cytokine which is expressed in the CNS during development but not profoundly expressed in the CNS of adults unless some pathological processes take place [114]. In particular, high levels of pleiotrophin and its respective receptor (protein tyrosine phosphatase receptor type z, PTPRz) are expressed in a subset of human glioblastomas; in general, pleiotrophin abundance is higher in human high-grade gliomas and is correlated with poor survival of patients [115]. The group of Anna Dimberg has shown that in a murine glioblastoma model, pleiotrophin activates the anaplastic lymphoma kinase 1 (ALK1), which acts as its receptor and leads to VEGF deposition and vascular abnormalities in gliomas. Finally, a more recent publication has also confirmed and expanded these findings since the pleiotrophin receptor ALK1 confers multiple advantages to glioblastoma cells through neovascularization and cell proliferation [116].

#### 3.1.6. EVs and miRs

The extracellular vesicles (EVs) are phospholipid bilayer-enclosed vesicles secreted by various cell types and broadly categorized to microvesicles (MVs up to 1000 nm in diameter) and exosomes (30–100 nm), depending on their size and origin (plasma membrane and endosomal, respectively) [117]. EVs are important mediators in brain tumors, playing a critical role in intercellular communication since their cargo is targeted to specific cell types where it exerts specific biological effects by reprogramming recipient cells and/or their surrounding microenvironment [118]. The cargo of EVs in brain tumors can be proteins, such as secreted factors discussed above but also microRNAs (miR). miRs form stem-loop structures and are transcribed first as longer premature RNAs which are then cleaved and processed by the Drosha and Dicer system into the final mature miR molecule, which interacts with a target complementary mRNA and silences it in the cytoplasm [119]. miR expression can be altered in cancer in a way that favors tumorigenesis and angiogenesis through a variety of mechanisms, including chromosomal changes, epigenetic defects, mutations, and alterations in the machinery involved in miR biogenesis [120].

Examples of miRs playing an important role in promoting angiogenesis in brain tumors are miR-19b and miR-9-5p [121,122]. The former has been found in glioblastoma-derived microvessels which also contain VEGF and PDGF. miR-19b further supports the pro-angiogenic role of the growth factors by repressing the expression of antiangiogenic proteins such as thrombospondin-1 [121]. The latter miR-9-5p has been very recently found in glioblastoma-derived EVs where it suppresses in vitro and in vivo the expression of genes like SOX7 and RGS5 which are responsible for maintaining physiological levels of angiogenesis and inhibiting EC growth, respectively [122].

### 3.2. Direct Interaction via Gap Junctions

The gap junctions are channels formed by the cell membrane proteins called connexins and they allow intercellular communication between adjacent cells [123]. One can easily understand that these gap junctions play an essential role in coordinating cellular processes not only under physiological settings but also under pathological settings such as cancer [124]. Different types of connexins can therefore possess either pro- or anti-tumorigenic functions [125].

When it comes to brain tumors, the most studied connexins are Cx43 and Cx30. In 2002, it was first shown that Cx43 expression enables glioma cells to interact not only with astrocytes but also with other cell types in the brain parenchyma, thus allowing a rapid invasion of the tumor into the brain [126]. In a subsequent study in 2003, it was shown that Cx43 is expressed both on EC and on glioblastoma cells. Direct cell-to-cell communication via Cx43 gap junctions seemed to be crucial for the transportation of VEGF from the glioblastoma to the EC and the promotion of tube formation in the latter [127]. More recently, in 2016, another study confirmed the importance of Cx43 in glioblastoma angiogenesis and also showed that a particular miR (miR-5096) is transported from the glioblastoma to the EC and further induces the expression of Cx43 [128]. Another connexin which plays a role in human gliomas is Cx30. Cx30 has an opposing role compared to Cx43 since it inhibits the growth of malignant cells [129]. A more recent study confirmed these findings and showed that Cx30 reverses the malignant phenotype of gliomas by modulating the expression of CD133, a molecule which, among other things, has a known pro-angiogenic role [130]. Although Cx30 seems to have a tumor suppressive effect in glioblastoma, we need to point out that it can also protect the glioma cells from radiation therapy [129]. Therefore, the potential of the pharmacological targeting of connexins needs to be interpreted cautiously.

### 3.3. Indirect Interaction via Intermediate Cells

The communication between the tumor and the endothelial cells is often regulated by other intermediate cells that reside in the tumor microenvironment and are part of the tumor’s milieu.

#### 3.3.1. Pericytes

Pericytes are another very important cell type supporting tumor angiogenesis and the pericyte–EC interaction is one of the most important ones for facilitating vessel formations in brain tumors [131]. Pericytes promote vascular maturation by expressing neuron-glial antigen 2 (NG2, also known as GSP4) and α-smooth muscle actin (α-SMA) [132]. Importantly, the interaction between the pericytes and the EC contributes to the formation of junctions between the EC and triggers signaling that leads to the recruitment of either resident or peripheral immune cells which can also support angiogenesis [133]. Pericytes are actually the equivalent of smooth muscle cells in other vessels and play a critical role in the regulation of vascular contractility and permeability [134]. Brain tumor development relies on fast and efficient vascularization and phenomena such as abnormal branching, disorganized networks, uneven basement membranes, and lack of pericytes favor non-productive angiogenesis and metastasis [135]. Therefore, delivery of pericyte therapy to the tumor can allow vascular normalization, preventing metastasis and improving the efficacy of chemotherapy or radiotherapy [136]. However, many tumor types rely on angiogenesis for growth, for which pericytes are critical. Thus, the inhibition of pericyte-induced angiogenesis may also be a promising therapeutic target to prevent tumor growth [136].

#### 3.3.2. Astrocytes

One main cell type of this milieu is the astrocyte, and the astrocyte–EC interaction is actually the main component of the BBB responsible for the formation of tight junctions between EC [137]. Astrocytes have been shown to be able to support tumor angiogenesis via multiple mechanisms that involve secretion of angiogenic and growth factors such as VEGF and protein carriers such as insulin and albumin [138]. Moreover, the astrocytes are capable of secreting endothelin-1 (ET-1), which has a potent mitogenic effect on EC apart from its classical role as a vasoconstrictor [139].

#### 3.3.3. Resident Microglial Cells and Tumor Associated Macrophages (TAMs)

The hypoxic microenvironment of the tumor and the disruption of the BBB observed in most brain malignancies facilitates the recruitment of immune cells such as peripheral macrophages and resident microglial cells to the tumor [131]. Both cell types secrete cytokines such as TGF-β and IL-6 that are known to maintain cancer cell stemness and support angiogenesis [131,140]. TAMs and tumor-associated microglia exhibit a phenotype known as M2 immunosupressive. The M2 phenotype not only helps the tumor cells to escape from tumor immunosurveillance, but M2 cells also secrete angiogenic factors like VEGF and MMPs that help tumor cell invasion [141]. These immune cells express a protein receptor called RAGE (receptor for advanced glycation end-products) which plays a key role in the stimulation of angiogenesis through TAMs and microglia. The interaction of RAGE with its ligand leads to downstream signaling that favors VEGF secretion and genetic disruption of RAGE signaling in mouse glioma models led to prolonged survival [142].

#### 3.3.4. Neurons

Despite the fact that neurons are abundantly present in the tumor microenvironment and the fact that glioma cells integrate into neural circuits, which are essential for glioma progression, this cell type is often overlooked [143]. In fact, so far, there is limited published work referring to the exact role of neurons in relation to angiogenesis in brain tumors. However, there is growing amount of evidence that neural cells can interact with the EC via secretory exosomes and microvesicles [144]. Moreover, neural stem cells which are precursor cells to both adult neurons and astrocytes secrete growth factors such as brain derived neurotrophic factor (BDNF), nerve growth factor (NGF), and neutrophin3 (NT3), which are all not only angiogenic but also facilitate EC survival and vessel stabilization [145].

## 4. Role of Hypoxia, Reactive Oxygen Species, Nitric Oxide, and the Systems that Produce Them

Hypoxia is a condition where a tissue has substantially lower levels of oxygen compared to what is considered normal for that specific tissue type [146]. Although the tissue oxygen tension (pO_2_) varies greatly within the normal CNS, it has been confirmed that the intra-tumoral tissue is consistently less oxygenated than the peritumoral areas [147]. Hypoxia supports tumor growth and expansion by using many different mechanisms with the HIF transcription factor family playing a core role in altering gene expression during hypoxia [146].

These mechanisms are listed below:Hypoxia is an essential environmental cue for the maintenance of the glioblastoma cell population [146].By lowering the amount of available oxygen, hypoxia impedes the formation of ROS from the mitochondria (discussed below) which have a well-known cytotoxic role. The formation of ROS is a hallmark cytotoxic mechanism in radiation therapy whose efficacy is thus significantly reduced in the presence of hypoxia [148].Hypoxia promotes genomic instability by attenuating the expression of DNA repair enzymes. Therefore, more mutations can be acquired in the DNA which eventually favor the tumor growth even more [149].Hypoxia promotes a switch of the cancer cell metabolism to aerobic glycolysis by amplifying glucose uptake and promoting conversion to lactate rather than acetyl-CoA. This aerobic glycolysis creates an intracellular environment where the TCA cycle and oxidative phosphorylation are attenuated and therefore there is an accumulation of glycolytic intermediates which can be then used for anabolic reactions and biosynthesis of nucleotides, phospholipids, and amino-acids which are all absolutely necessary for tumor cell growth and proliferation [150].Last but not least, hypoxia supports and favors many of the different mechanisms for new blood vessel formation discussed above such as sprouting angiogenesis, recruitment of EC progenitor cells from the bone marrow and vasculogenic mimicry. In particular, hypoxia stabilizes the HIF-1 family of transcription factors which promote the expression of genes such as VEGF, SDF-1, and angiopoietin from the glioblastoma stem cells [58,151,152].

Reactive oxygen species (ROS) are chemicals which belong to two main categories: (a) free radicals (e.g., superoxide, hydroxyl radical) and (b) non-radicals (e.g., hydrogen peroxide). These small molecules serve as signaling messengers when they are produced at low levels and have crucial physiological roles in host defense, hormone biosynthesis, synaptic plasticity, etc. However, it is also well known that higher levels of ROS increase overall oxidative stress and contribute to the progression and the pathology of several diseases such as cancer [153,154].

The accumulation of ROS inside the cell has been associated with the progression of gliomagenesis on many different levels. First, ROS can induce direct DNA damage since molecules such as the hydroxyl radical or hydrogen peroxide can lead to the suppression of tumor suppressor genes like p53 or the activation of oncogenes such as Ras [155]. In particular, ROS-induced mutations on p53 can lead to dysregulation of neoangiogenesis and the progression from low- to high-grade gliomas via transcriptional activation of VEGF and FGF expression [156,157,158]. Second, ROS can lead to the activation of membrane receptors with key role in the regulation of the intracellular signaling such as the EGFR and the PDGFR. The activation of these receptors can in turn activate tyrosine kinases and related downstream pathways such as the prosurvival pathways of PI3k/Akt and extracellular signal-regulated kinases (ERKs) which play a very important role in the progression of glioblastoma [155,159].

Mitochondria, xanthine oxidoreductase (XOR), NADPH oxidases (NOX), and uncoupled endothelial NO synthase (eNOS) are considered the major cellular sources of ROS [160,161,162,163], which have been shown to play a different regulatory role in brain tumorigenesis [164]. Among all the different isoforms of NOX, NOX4 has been shown to have the most crucial role in glioblastoma, and NOX4-derived ROS (like H_2_O_2_) are essential for glioblastoma invasion and for angiogenesis around the tumor [165]. Therefore, and according to this study, NOX4 inhibition with lentivirus shRNA could be an attractive therapeutic option for overcoming radioresistance in glioblastoma. Moreover, recent data from our group have further expanded our knowledge around NOX4 and indicate that TGF-β or PDGF can activate NOX4-mediated ROS production which in turn activates the nuclear factor erythroid 2 (NFE2)-related factor 2 (Nrf-2)-mediated transcription and leads to higher proliferation and self-renewal of human glioma stem cells [166].

Mitochondria are another major source of ROS and enhanced production of mitochondrial-derived superoxide anion (O_2_^−^), has been shown to trigger genetic instability and inflammation in glioblastoma [167]. These processes can initiate dedifferentiation processes of astrocytes to astrocytoma cells and further increase the tumorigenic potential of these cells. However, it is well-known that mitochondrial-derived ROS can trigger cytotoxicity and apoptotic cell death; therefore, several pharmacological reagents that aim to increase mitochondrial ROS have been tested in glioblastoma and they have been shown to successfully increase the sensitivity of glioblastoma to chemotherapy and radiotherapy [168,169].

Apart from mitochondria and NOX, another major source of ROS which has not been thoroughly studied in glioblastoma is the enzyme xanthine oxidoreductase (XOR). XOR is an enzyme controlling purine catabolite that simultaneously produces ROS (such as O_2_^-^ and H_2_O_2_) or nitric oxide (NO) depending on the cellular conditions [170]. Very little is known about the role of XOR in glioblastoma despite the fact that it has been shown that the activity of this enzyme is largely increased in brain tumors compared to normal tissue [171]. Moreover, it has been shown that XOR is essential for EC survival and promotes angiogenesis, but no such studies have been performed in relation to brain tumors [172]. To the best of our knowledge, there is only one pharmacological study available where the authors used a natural antioxidant called ACA in glioblastoma cells. This antioxidant can also act as a XOR inhibitor and is able to inhibit proliferation and migration of glioblastoma cells in vitro [173]. Unfortunately, studies with more selective pharmacological inhibitors of XOR such as allopurinol or febuxostat are currently missing, although this could open new therapeutic paths in the battle against glioblastoma.

Finally, an enzyme which has been involved in ROS production and the induction of oxidative stress is the uncoupled form of eNOS which produces O_2_^-^ instead of NO, and it further promotes tumorigenesis by creating a pro-oxidant environment [174]. eNOS uncoupling has not been studied so far in brain tumors and it might actually give rise to an interesting new research field.

NO is a small signaling molecule belonging to the family of gasotransmitters and has many important pathophysiological actions. It is generated in a vast number of different cell types by three different NOS isoforms which catalyze a complex five electron oxidation of the amino acid L-arginine to form NO and L-citrulline. Continuous NO production is carried out by the two constitutively expressed NOSs, the endothelial (eNOS, NOS 3) and the neuronal (nNOS, NOS 1), which produce moderate amounts of NO under physiological conditions [175]. The third NOS isoform, inducible (i) NOS (NOS 1), was first discovered in macrophages but later found to be expressed in various cell types [176]. Upon induction (for example, during inflammation), iNOS produces large amounts of NO that can react with superoxide to form peroxynitrite. This iNOS-derived NO has important functions in the innate immune response. However, large quantities of peroxynitrite or chronically elevated peroxynitrite levels can also be harmful and aid in the progression of several pathological conditions [177].

When it comes to the role of different NOS isoforms in different human brain tumors from clinical specimens, it seems that the two main NOS isoforms expressed in glioblastoma cells are nNOS and eNOS, whereas the endothelial cells of the surrounding vessels express eNOS [178,179,180]. Some of these studies also showed that there is a positive correlation between the expression levels of NOS and the expression of VEGF and its receptor VEGFR-1 indicating a possible link between NO synthases and the induction of angiogenesis in brain tumors [179,181]. The findings on human samples have been also confirmed in in vivo studies, like in a rat model of glioblastoma where eNOS seemed to play a central role not only in angiogenesis but also in the disruption of the BBB and the creation of peritumoral edema [182]. Finally, a hallmark study published in 2011 showed that iNOS is surprisingly expressed in glioblastoma stem cells and induces stem cell proliferation and tumor growth which both were blocked in vivo after administration of an iNOS inhibitor [183].

Overall, the current literature indicates all three NO synthases as being upregulated in brain tumors and supporting tumor growth and progression mechanisms. However, one limitation of several of the available studies is that they are quite descriptive; therefore, future studies which could possibly measure more accurately the NO levels in different brain tumor segments are needed. Furthermore, analyzing whether there is any interaction between ROS and the formation of nitrosative stress, apart from the confirmed presence of oxidative stress in brain tumors, is highly interesting. The roles of ROS, NO, and of the systems producing them are summarized in Figure 3 below.

## 5. Conventional and New Interventions that Target the Endothelium in Brain and CNS Tumors

The first-line conventional treatments for glioblastoma and other CNS malignancies are surgical resection, radiotherapy, and chemotherapy. Despite that these treatments do not have the endothelium as a primary target, they do have multiple impacts on the brain’s vasculature which must be therefore taken into consideration before administrating a newer intervention that targets the endothelium *per se*.

The surgical resection of the brain tumors is often overlooked or taken for granted since there is an increasing amount of new available therapies that target the glioblastoma and other CNS malignancies. However, surgery still remains the most important first step in the treatment for brain and CNS malignancies, and it is a prerequisite for a prolonged survival [184]. Injury of the cerebral vasculature after surgical resection of the tumor is a rare (1–2%) but very devasting complication [185]. The major vascular complication after surgery is called cerebral vasospasm and is associated with high morbidity and mortality in the reported cases (~30%) [186]. The cerebral vasospasm implies a narrowing of the blood vessels resulting in decreased blood flow and eventually ischemia to distal tissues in the brain [187].

Apart from the surgical resection, radiation therapy is also an absolutely necessary post-operative approach in glioblastoma, as it induces ROS production which leads to cancer cell death and prolongation of the patients’ survival [188]. Despite being so effective in the initial treatment phase, radiation therapy has been implicated in triggering recurrence mechanisms and relapse [189]. One of the main mechanisms involved in that process is the radiation-induced vascular remodeling, since radiation affects vascular integrity causing vasculopathy, vascular depletion, hypoxia, and neo-angiogenesis [190,191]. Moreover, radiation induces changes in cell density, loosens the tight junctions, and therefore increases the blood–brain-barrier (BBB) dysfunctionality and permeability leading to higher infiltration of immune cells and BBB leakage [192,193].

When it comes to chemotherapy, temozolomide (TMZ) is the most commonly used anti-glioma chemotherapeutic agent which acts by alkylating guanine residues and inducing DNA damage and cell cycle arrest [194]. Despite the usefulness and cytotoxicity of TMZ in glioblastoma cells, this molecule has also been associated with mechanisms that could favor tumor growth and relapse [195]. For example, a recent study indicated that chemotherapeutic stress induced by TMZ leads to trans-differentiation of glioblastoma cells to endothelial-like cells and promotes vasculogenic mimicry (VM) [196]. It is also important to mention that according to another study, although TMZ has a very good cytotoxic effect on the glioblastoma cells *per se*, it has almost no effect on glioblastoma-associated EC and is unable to reduce the peritumoral vascular density *in vivo* [197]. Considering the lack of TMZ in the vasculature or, even worse, the fact that it can induce VM mechanisms, it becomes evident that TMZ should not be administrated alone but rather combined with antiangiogenic drugs and radiotherapy [194].

As discussed above, brain tumors are among the most vascularized solid tumors found in humans, and blood vessels play a key role in supporting tumor progression. Therefore, several antiangiogenic therapies have been tried so far with unfortunately limited or no improvement in overall survival (OS) [198]. The only FDA approved drug since 2009 is bevacizumab (Avastin), which is a human monoclonal antibody that neutralizes VEGF-A activity and thus shows antiangiogenic action [199]. Several clinical studies have been conducted so far and the conclusion is that, despite the fact that Avastin improves significantly progression-free survival (PFS) for six months, it does not improve OS. The failure of this antibody has multiple explanations with the most important one being that VEGF is not the only growth factor regulating angiogenesis in brain tumors.

As discussed earlier, most of the brain and CNS tumors and especially glioblastoma are characterized by excessive levels of hypoxia which is one of the main reasons leading to a reduced efficacy of the antiangiogenic drug bevacizumab (Avastin) [146]. The molecular mechanisms behind this inadequacy of Avastin implicate, for example, hypoxia-mediated upregulation of the gene HIG2 or downregulation of the gene CYLD [200,201]. The HIG2 gene encodes for a protein which correlates with the tumor’s grade, is associated with poor prognosis, and induces higher HIF-1β, VEGF expression, and resistance to bevacizumab [201]. On the other hand, when the gene CYLD is suppressed by hypoxia, this leads to excessive inflammation and is possibly linked with a reduced long-term efficacy of Avastin [200]. Overall, it becomes evident that apart from the classic anti-VEGF therapy with Avastin, it is important to administrate in the patients’ complementary substances that are either resistant or activated by hypoxia and then exert a cytotoxic effect or drugs that target directly critical molecular mediators of hypoxia such as the HIF transcription factor family [202,203,204]. Such drugs are, for example, the molecule TH-302, which is activated under low oxygen tension and has a cytotoxic effect [203], or the molecules amphotericin-B and 2-methoxyestradiol, which have a HIF inhibitory activity [202,204].

Since then, additional efforts have been made with small molecule kinase inhibitors that target multiple receptors involved in angiogenesis in glioblastoma and other brain tumors such as PDGF-R, FGF-R, VEGF-R, etc. These molecules have been so far used in Phase I–III trials; however, all of them have also failed and they are actually inferior to Avastin [205].

The brain tumors are not an exception to the general concept in cancer stating that the cancer cells find many times a way to develop resistance to anti-cancer drugs and acquire new properties. The challenges are even greater in brain tumors considering the significant amount of brain edema which greatly increases the morbidity and mortality and the obstacles for drug delivery posed by the BBB [206]. Some relatively new efforts have been made which actually aim to target completely new pathways. Two examples are the targeting of the renin angiotensin system (RAS) and the angiopoietin-2 receptors system (Ang-2 R).

Angiotensin II receptors (AngII-R) have been found to be expressed not only in the glioblastoma stem cells but also in EC, and their activation promotes tumor cell proliferation and angiogenesis [207]. Recent studies indicate that the use of ASIs (angiotensin system inhibitors) is associated with longer OS in both newly diagnosed and recurrent glioma patients in combination with chemotherapy and/or Avastin [208]. Another possibly promising therapeutic regime is the dual inhibition of Ang-2 R and VEGFR. Ang-2 plays a significant pro-angiogenic and immunomodulatory role in brain tumors [209]. According to two publications published by two independent groups in 2016, the combined targeting of Ang-2 and VEGF inhibits tumor growth and improves overall survival via targeting not only the EC but also the surrounding TAMs and the resident microglia [210,211]. These two studies were conducted in pre-clinical murine models and therefore future studies in humans are warranted.

## 6. Conclusions and Future Prospects

Overall, it becomes evident that since the progression of brain tumors and especially glioblastoma is a multi-staged and very complex process, the future therapeutic approaches should include combinatorial therapy that targets most of the involved cell types in the cancer milieu alongside with the respective signaling pathways. Glioblastoma and other brain tumors are highly vascularized and among the ones with the highest morbidity and mortality so far. The deeper understanding of the interaction between EC surrounding the tumor blood vessels and the tumor cells will hopefully reveal new molecular pathways and pharmaceutic targets and therefore improve the quality of life and the survival of the patients suffering from brain malignancies.

## Figures and Tables

**Figure 1 ijms-21-07371-f001:**
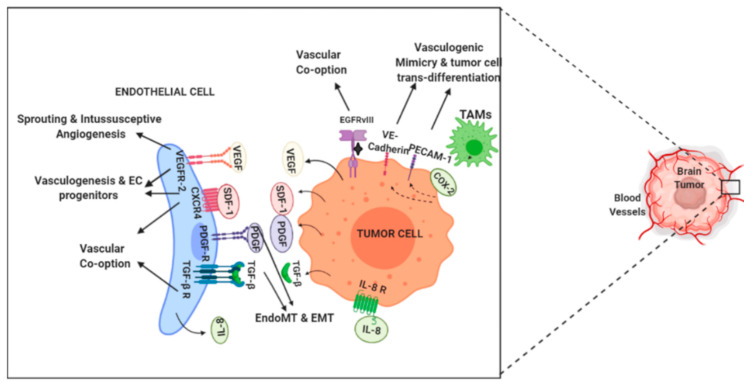
Mechanisms and molecular mediators involved in formation of new vessels that promote further tumorigenesis in brain and CNS tumors. Secreted factors from the tumor cells such as VEGF, SDF-1, PDFG, and TGF-β act upon respective receptors on the endothelial cells and trigger new vessel formation mechanisms such as sprouting and intussusceptive angiogenesis, vasculogenesis, vascular co-option, and recruitment of endothelial cells (EC) from the bone marrow. PDGF and TGF-β also trigger mechanisms such as EndoMT (on the endothelial cells) and EMT (on nearby epithelial cells) which also further promote angiogenesis. There is also a positive feedback loop mechanism on TGF-β signaling with the EC-derived IL-8 acting as the mediator. Moreover, proteins expressed in the tumor cells such as VE-Cadherin, platelet endothelial cell adhesion molecule-1 (PECAM-1), and COX-2 (activated by nearby tumor associated macrophages—TAMs) facilitate mechanisms like the vasculogenic mimicry and the trans-differentiation of the cancer cells to endothelial cell-like phenotypes. Finally, a mutated (and constantly activated) version of EGFR (EGFRvIII) on the tumor cells can also support the mechanism of vascular co-option.

**Figure 2 ijms-21-07371-f002:**
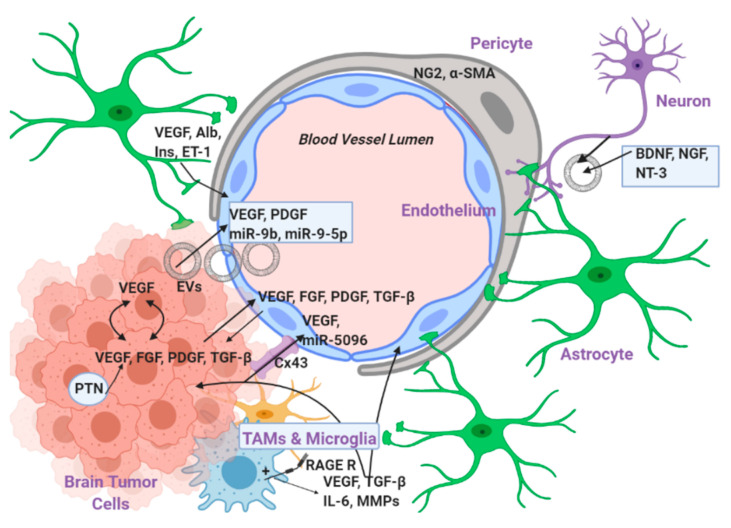
Levels of interaction between the endothelial and the tumor cells. The interaction happens on three different levels: (a) secreted factors, (b) gap junctions, and (c) intermediate cells. The secreted factors are derived either from the tumor or the endothelial cells and there is a reciprocal relationship between them. (a) Brain tumor cells produce and secrete VEGF, FGF, PDGF, and TGF-β that act upon the endothelial cells and induce angiogenesis with mechanisms discussed in Section 2. Moreover, the tumor cells secrete extracellular vesicles (EVs) that contain secreted protein factors and microRNAs that are also positively correlated with the induction of angiogenesis. Note also that there is a positive feedback loop mechanism between pleiotrophin (PTN) and VEGF production on the tumor cells. (b) There is also direct interaction between the tumor and endothelial cells in the brain via connexins such Cx43. Cx43 facilitates the transportation of microRNAs and proteins like VEGF from the tumor to the endothelial cells and therefore is directly involved in the induction of angiogenesis. (c) Many types of surrounding intermediate cells also play a crucial role in the communication between the endothelial and the tumor cells. Pericytes promote vascular maturation by expressing NG2 and α-SMA. It is important to note that the interaction between the pericytes and the endothelial cells contributes to the formation of junctions between the endothelial cells and triggers signaling that leads to the recruitment of either resident or peripheral immune cells, which can also support angiogenesis (see discussion about TAMs below). Astrocytes secrete factors that promote and support angiogenesis and vessels growth such as VEGF, albumin (Alb), insulin (Ins) and endothelin-1 (ET-1). Additionally, the resident microglia and the tumor associated macrophages (TAMs) which reside locally and have an M2 tumor-promoting phenotype play a very important role in supporting tumorigenesis and angiogenesis via their receptor RAGE1, whose activation triggers the secretion of VEGF, matrix metalloproteinases (MMPs), TGF-β, and IL-6. Finally, the surrounding tumor neurons seem to also affect the endothelial–tumor cell interaction since they secrete factors such as BDNF, NGF, and NT3 which are all not only angiogenic but also facilitate endothelial cell survival and vessel stabilization.

**Figure 3 ijms-21-07371-f003:**
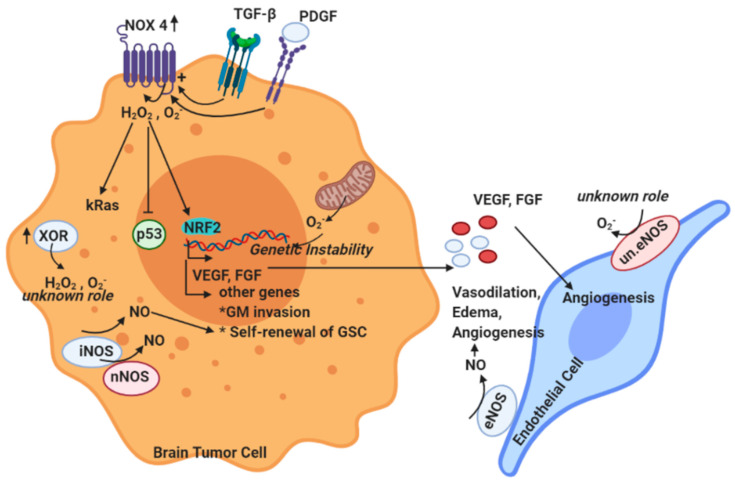
Summary of the ROS and NO producing systems expressed in the tumor and endothelial cell respective and their role in tumorigenesis and angiogenesis. The main ROS producing systems in the tumor cells are the NADPH oxidases and especially NOX4, the enzyme XOR, and the mitochondria. NOX4 activation is positively correlated with the induction of tumorigenic and angiogenic mechanisms. In particular, secreted factors acting on the tumor cells such as TGF-β and PDGF activate NOX4, which in turn produces high amounts of ROS such as H_2_O_2_ and O_2_^-^. These ROS alter the downstream signaling in a way that promotes the expression of pro-angiogenic genes such as VEGF and FGF, which are then secreted and act on the endothelial cells. Moreover, NOX4-derived ROS induce the expression of other genes involved in glioblastoma invasion or in processes such as the self-renewal of glioblastoma stem cells (GSC). Note that in all these processes, the activation of the transcription factor Nrf-2 plays a key role. Mitochondria can produce large amount of ROS that can lead to genetic instability and inflammation which can facilitate the dedifferentiation of astrocytes to astrocytoma cells and increase their tumorigenic potential. However, the mitochondrial-derived ROS can play also opposing roles and facilitate the apoptotic cell death of tumor cells. The main NO is βeNOS, which is among the master regulators of the vascular tone (inducing vasodilation). iNOS-derived NO can trigger the self-renewal of GSC, whereas NO from other sources (i.e., nNOS and eNOS) has been positively correlated to phenomena such as vasodilation, edema, and angiogenesis. Finally, it is important to mention that the phenomenon of eNOS uncoupling (un. eNOS), when eNOS switches activity from a NO to a ROS producing enzyme, has not been studied at all in the context of brain tumors.

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
