# Peer review of "Endothelial-Tumor Cell Interaction in Brain and CNS Malignancies"

_ijms, 2020, doi:10.3390/ijms21197371_

Round 1

Reviewer 1 Report

The review is generally well written and gives a fairly comprehensive view of the dependencies of tumor on its vascular environment.

I would suggest that the authors should include a note on Hypoxia and its role in angiogenesis and GBM progression, since hypoxia is currently a considerable challenge in the path of GBM treatment and patient prognosis. Furthermore, please add a note on how hypoxia impacts Avastin's efficacy. 

The authors also need to discuss standard-of-care therapies employed in brain tumors - surgery, radiation and Temozolomide and how these impact the brain vasculature, compromise the blood brain barrier and impact the vascular microenvironment of the tumor. These are clinically relevant discussions that will inform the reader of the current status of the field and where we are behind. 

The authors also need to revise the manuscript and correct awkward sentence structures and grammatical errors. For example:

  1. Line 132: "Cues" instead of "Queues" 
  2. Lines 166, 188, 196, 340, 615: awkward sentences/grammar

Author Response

Dear reviewer,

You can find a point-by-point answer to your comments in the attached word document.

Reviewer 2 Report

This excellent manuscript gives a good overview on the topic of angiogenesis in brain tumors and especially glioblastomas. The different mechanisms of angiogenesis are explained in details. The linguistic style is very good, which makes the manuscript easy to read and understand. All relevant literature is cited. In summary, I strongly recomend publication of the manuscript without further changes needed.

Author Response

Thank you for your nice comments and for finding our work so interesting!

Round 2

Reviewer 1 Report

Other than a few grammatical edits, the authors have vastly improved the manuscript.